# The Impact of Informal Social Support on Older Health: Evidence from China

**DOI:** 10.3390/ijerph19042444

**Published:** 2022-02-20

**Authors:** Daisheng Tang, Xinyuan Wang, Yahong Liu, Tao Bu

**Affiliations:** School of Economics and Management, Beijing Jiaotong University, Beijing 100044, China; dstang@bjtu.edu.cn (D.T.); 14120553@bjtu.edu.cn (Y.L.)

**Keywords:** informal social support, formal social support, older health, healthy aging

## Abstract

Objectives: To explore the impact of informal social support (ISS) on older health. Methods: Multiple regression was used as the baseline regression, grouping regression was used to examine whether there were health effect differences among groups based on age and household registration, and insurance was selected to explore moderating effects of formal social support (FSS). Results: First, economic support, accompanied support, and number of intimate contacts had significantly positive effects on older health except for care support’s negative effects. Second, ISS had different health effects for different groups based on age and household registration. Third, FSS was a significant moderating for ISS. Conclusions: The government should emphasize and strengthen the supplementary role of ISS to FSS and promote the effective combination of the two, especially for the older who are high-age and rural, and further improve the role of care support.

## 1. Introduction

Population aging is an inevitable global trend. The United Nations stipulates that the countries and regions where more than 10% of the population is over 60 years old will become aging societies. In China, the population aged 60 and higher was 253.88 million in 2019, accounting for 18.1% of China’s total population (National Bureau of Statistics 2020), which far exceeds the international standards of aging. Furthermore, the proportion of the older population is still rising. As the average age of society continues to increase and society grows more developed, people are beginning to pay greater attention to older health. In addition, according to numerous studies, health is strongly related to professional activity [1,2]. According to China’s current retirement policy, the retirement ages for most males and females in China are 60 and 55, respectively, which also poses a challenge to China’s social security system. However, China has not yet developed adequate social security capabilities, which is forcing people to turn to other paths for assistance with healthy aging. Through Chinese cultural traditions that value family, informal social support (ISS) can provide various services to the older, especially while formal social security capacities remain insufficient; ISS provided by family members as the core has become a major determinant of older health.

Social support is a key concept of social gerontology and is closely related to older health [3]. Social support can be divided into formal social support (FSS) and ISS, depending on who is providing the support. FSS refers to social security policies and medical security systems provided by governments and communities at all levels; ISS refers to economic support, daily care, and emotional support provided by family members, friends, or neighbors [4,5]. In the study of ISS for black people, in addition to the support provided by family relatives and friends, the church is also a major source of ISS [6,7,8].

Scholars who view the role of facilitation believe that ISS can not only make up for the shortcomings of social security systems [9,10], but also replace some of the professional care with the care of family members, relatives, and friends, which can reduce the cost of nursing for the elderly [11]. This replacement is beneficial to elderly health thus informal care support can be a useful supplement to professional care [12]. An analysis of 695 elderly men and women conducted by Choi N.G. and Wodarski J.S. (1996) found that emotional and instrumental aid from spouses, children, and other relatives appeared to contribute significantly to the prevention of further physical and functional deterioration [13]. Phillips D.R. et al. (2008) used data from interviews with 518 older people over 60 years old in Hong Kong to discuss the importance and effectiveness of informal support for the mental health of the older. They found that informal support from family members, relatives, friends, or neighbors can effectively enhance older mental health and that the impact of family support is the most significant [14]. Wu H. and Lu N. (2017) used data from the 2011–2012 China Health and Retirement Longitudinal Study (CHARLS) to discuss the impact of children’s informal care on the health behaviors of elderly people with chronic diseases through the propensity score matching method and found that children’s informal care can improve the health behaviors of the elderly and thus improve elderly health [15]. 

However, scholars who hold an inhibitory point of view, starting from the self-perception of support providers or recipients, believe that ISS is not conducive to improving elderly health. From the perspective of the support provider, the provision of ISS will place varying degrees of burdens on the providers [16]. This kind of burden puts support providers under physical and psychological stress, which has a negative impact on their health, which in turn affects elderly health [16,17]. Therefore, caregivers can be connected to the community [17] to develop community home care for the older, or increase the number of caregivers or the duration of professional care [18] to share care responsibilities and stress. From the perspective of the recipient of support, the family’s support will place a certain psychological burden on the older [19], thereby reducing the older’s sense of self-efficacy [20] and negatively affecting their physical and mental health. Zhang W.J. and Li S.Z. (2004) used the Data Collection of the Healthy Longevity Survey in China, completed in 2000, to explore the impact of children’s intergenerational support on the physical and mental health of elderly seniors, and they found that living with children and receiving economic support from children has a negative impact on the elderly’s ability to take care of themselves [21]. Wang P. et al. (2017) used a hierarchical linear model to explore the impact of intergenerational support on the mental health of rural elderly. They found that receiving the help of children in completing housework will cause a decline in the mental health of the elderly [22].

Previous studies have focused on the impact of ISS on older health, and there is a certain degree of controversy about this issue. However, most of these studies concentrate on how support from family, especially from children, impacts older health. This actually narrows the concept of ISS whose sources include family members, relatives, friends, neighbors. Attaching importance to blood relationships and geo-relationships creates a unique community of human sentiment in China [23], leading to the spread of family-centric networks that are also important for Chinese people. In China, this unique cultural tradition has been widely accepted by people and continues to this day, which gives ISS a more special role for the Chinese. Therefore, we have the reason that ISS has positive health effects on Chinese older people, which this paper seeks to verify. In addition, based on the Chinese context, China’s long-standing urban-rural gap and changes in the fertility policy should be taken into consideration, which will cause the impacts of ISS to vary among different groups of people.

Therefore, the objective of this study was to explore the impact of ISS on older health by selecting comprehensive indicators of ISS and older health through the use of a multiple linear regression model. Furthermore, we used grouping regression to explore whether there are divergences in the health effects of ISS based on age and household registration. In addition, we selected insurance as a moderating variable to explore the moderating effect of FSS on ISS.

## 2. Methods

### 2.1. Data and Sample

The data used in this paper came from a sample survey of the Aging Strategy Study in Chengdu, Sichuan Province, China, in 2019. This cross-sectional survey, carried out in 2019 by the Health Commission of Sichuan Province, provided valuable information on the social support and health status of older people. In recent years, Chengdu has achieved remarkable results in healthy aging through the combination of medical care and older care. According to Chengdu Daily (2020), the proportion of the older (60 years old and above) in Chengdu in 2019 was 21.07%, and the average life expectancy was 81.01 [24], which were higher than the national aging population proportion of 18.1% and the average life expectancy of 77.10. According to the relevant data published by the National Bureau of Statistics and National Health Commission of the People’s Republic of China [25], in the overall rankings of the average life expectancy of Chinese provinces in 2019, Shanghai ranked first (80.26), which was lower than Chengdu’s average life expectancy (81.01). This high life expectancy of the population indicated that Chengdu was doing a good job in offering healthy aging and that conducting relevant studies on older health in Chengdu could provide a reference for the national aging strategy. Therefore, it was reasonable to select data from Chengdu in this paper. The survey subjects were middle-aged and older people (Chinese citizens aged 45 and above) living in the administrative area of Chengdu, and 767 samples were collected. The questionnaire consisted of 8 parts, which included: the basic situation; family situation; living habits, health conditions, and medical and health behaviors; the demand for aging health supplies and services; mental and cultural status; living environment; social participation status; the degree of support of external environmental resources for the healthy aging of respondents. The research object of this paper was the older aged 60 and over in Chengdu. Based on the original sample data, we finally obtained 449 valid samples after excluding groups under 60 and deleting and filling in the missing values.

### 2.2. Variables

#### 2.2.1. Older Health 

In 1989, WHO reaffirmed the definition of health to include 4 aspects: “physical health, mental health, good social adjustment, and moral health,” which had been widely recognized in academic circles. Taking the World Health Organization’s definition of health as a guideline, scholars had studied older health from self-rated health [26,27,28], physical health such as ADL and IADL [29], mental health such as degree of depression [30,31], and life satisfaction [32]. 

Self-rated health was an expression of social, psychological, and biological dimensions [33], which could reflect the physical and mental health status of individuals to a certain extent [34,35]. Therefore, this paper selected self-rated health to represent older health. It was phrased in the questionnaire as “How do you feel about your health?” The response was “very good, relatively good, fair, relatively poor, very poor”; we assigned the values 5, 4, 3, 2, 1 in order.

#### 2.2.2. Informal Social Support 

Social support was a key concept of social gerontology and was closely related to older health [3]. It could be divided into 2 aspects: FSS and ISS. FSS referred to social security policies and medical security systems provided by governments and communities at all levels, ISS referred to economic support, daily care, and emotional support provided by family members, friends, and neighbors [4,5]. ISS variables could also be divided into the categories of instrumental support and emotional support [36]. In this paper, the instrumental support category included care support and economic support; the emotional support category included accompanied support and number of intimate contacts.

*Care support.* It included 3 aspects, which were the care support from the son, daughter, and spouse, respectively. The questionnaire asked, “who is your primary caregiver?” Responses included “others, spouse, daughter, son”. We defined “others = 1, spouse = 2, daughter = 3, son = 4”. We divided it into 3 dummy variables according to the difference of the primary caregiver, which were son’s care support, daughter’s care support, and spouse’s care support. If the son was the primary caregiver of the older, it would be assigned a value of 1; otherwise, the value would be 0. The other 2 variables were also defined this way.

*Economic support.* This was expressed as “the proportion of medical expenses paid by children or others for the older for the self-paid part of care”. This was a continuous variable presented in the form of a percentage. The higher the value, the greater the economic support for the older.

*Accompanied support.* This was whether the older lived with family members. Living with the older could provide them with some emotional comfort. It was phrased in the questionnaire as “Who lives with you: children, spouse, others?” The response of “children” or “spouse” was defined as living with family members and was assigned a value of 1; otherwise, the value was 0.

*Number of intimate contacts.* This represented the older people’s social network to some extent. Frequent interactions with relatives, friends, or neighbors could alleviate loneliness in old age and provide spiritual comfort, thus the number of intimate contacts was classified as emotional support. The questionnaire phrased this as “How many friends do you have close contact with in a year?” The response was a continuous variable, which reflected the number of relatives and friends outside the family who had close contact with the older throughout the year.

#### 2.2.3. Moderating Variables

Considering the health effects of formal social support, we selected insurance as a moderating variable to explore the mediating effects of FSS in the process of ISS affecting older health. It was expressed in the questionnaire as “Do you have insurance?” “Yes” indicated that the older had insurance and was assigned a value of 1; otherwise, the value was 0.

#### 2.2.4. Control Variables

Control variables included personal characteristics (age, gender, marital status, child), socio-economic characteristics (education, income, insurance), and lifestyle characteristics (smoke, drink, exercise). Age was the actual age of the respondent. Marital status was dichotomized into the responses “with spouse = 1” and “without spouse = 0”. Child was defined the number of children the older had. Education was classified into 8 categories: not attended school (including not attended school = 1, primary school and below = 2, middle and high school = 3, University and above = 4. Income was expressed in the questionnaire as “How much is your monthly income: low (less than 5000 yuan) = 1, medium (5000–10,000 yuan) = 2, high (above 10,000 yuan) = 3”. The definition of Insurance was the same as the Section 2.2.3. Smoke was dichotomized into the responses “smoking = 1” and “non-smoking = 0”. Drink was dichotomized into the responses “drink = 1” and “non-drink = 0”. Exercise was expressed as “How often do you exercise in a week: never exercise = 1, 1 to 2 times = 2, 3 to 5 times = 3, 6 times and above = 4”.

### 2.3. Statistical Analysis

Stata (version 16.0, StataCorp, Computer Resource Center, College Station, TX, USA) was used for data analysis. Continuous variables (economic support, number of intimate contacts, age, child) were expressed as mean, standard error, median, maximum and minimum, and categorical variables (self-rated health, son’s care support, daughter’s care support, spouse’s care support, accompanied support, gender, marital status, education, income, insurance, smoke, drink, and exercise) were expressed as number, percentage, median, maximum, and minimum. In addition, this paper also calculated the Intergroup differences to indicate significantly different groups with estimated marginal means and standard errors. Then, a multiple linear regression model was used as the baseline regression to explore the impact of ISS on older health; ISS served as the independent variable and older health as the dependent variable. Furthermore, we divided the samples into groups according to age and household registration to explore the health effects of ISS on different groups. In this study, statistical significance was set at *p* < 0.05.

## 3. Results

### 3.1. Characteristics of the Participants

We identified 449 participants as analytical subjects based on the age criteria of 60 and above. Table 1 shows the descriptive statistical results of the variables involved in this paper. For older health, the proportion of the health status was very good, relatively good, fair, relatively poor, poor were 7.35%, 35.63%, 43.65%, 11.80%, and 1.56%, respectively. For ISS, the primary caregiver of the older were son, daughter, and spouse were 125, 39, 180; children or others bore most of the medical expenses of the older with an average rate of up to 81.1%; 86.2% of the older live with their families; the average number of intimate contacts made in a year were 12. For control variables, the average age of the participants was 76, of which 55.9% were male and 65.3% had spouses; the mean value of the number of children were 2.5; the proportion not attending school, primary school, middle and high school, university and above were 19.38%, 42.32%, 31.85%, 6.46%, and the proportion of low income, medium income, and high income were 79.29%, 15.37%, 5.35%, respectively; the percentage of smoking and drinking were 85.08% and 82.63%, respectively; the older exercised about twice a week on average. In addition, 99.1% of the older had insurance indicating wide coverage of formal support. Furthermore, this paper also calculated the Intergroup differences of older health. Table 1 showed older health had significant differences between the different groups.

### 3.2. Baseline Results

Table 2 shows the baseline results of the impact of ISS on older health. Model 1 explored the impact of son’s care support, daughter’s care support, spouse’s care support, economic support, accompanied support, and the number of intimate contacts on older health. Model 1 showed that son’s care support (*p* < 0.001), daughter’s care support (*p* < 0.001), spouse’s care support (*p* < 0.01) all had significant negative effects on older health; accompanied support (*p* < 0.01) and the number of intimate contacts (*p* < 0.01) had a significantly positive effect on older health, but economic support had no significant impact on older health. Models 2 to 4 gradually introduced three types of control variables: personal characteristics, socio-economic characteristics, and lifestyle characteristics based on Model 1. The absolute value of the son’s care support coefficient decreased from 0.565 to 0.449, the absolute value of the daughter’s care support coefficient decreased from 0.600 to 0.572, the absolute value of the spouse’s care support coefficient decreased from 0.313 to 0.288, which indicated that the negative impact of care support from the son, daughter, and spouse was gradually weakening with the addition of various control variables. Moreover, the coefficient of economic support increased from 0.270 to 0.382 and became significant starting from Model 3, the coefficient of accompanied support decreased from 0.299 to 0.221 in Model 3 and then increased to 0.274 in Model 4, which indicated that the positive effects of economic and accompanying support on older health were strengthening with the addition of various control variables. The role of ISS was becoming more and more important as the age of the older increased and economic income decreased, etc. In addition, the coefficient of the number of intimate contacts barely changed. During this process, the suitability of fit gradually increased, and the conclusions remained robust.

Control variables could be seen from Model 4 in Table 2. Education (*p* < 0.01) and exercise (*p* < 0.001) had significant positive effects on older health. 

### 3.3. Heterogeneity Analysis

Considering that ISS had different health effects on older people depending on their personal conditions, we attempted to explore the heterogeneity of the impact of ISS on older health based on age and household registration within the Chinese context. 

#### 3.3.1. Heterogeneity Analysis by Age

Under most circumstances, individual health would gradually decline with age. Therefore, we divided the sample into two groups according to age: low-age (60–79 years old), and high-age (80 years old and above). The regression results were shown in Models 5 and 6 in Table 3. Son’s care support had a significantly negative effect on the low-age group, and care support from the daughter had significantly negative effects on the high-age group. Economic support and accompanied support both had positive effects on older health within the high-age group. 

#### 3.3.2. Heterogeneity Analysis by Household Registration

The Chinese household registration system classified each member of the population as having an agricultural (rural) or non-agricultural (urban) status (hukou), with a sharp differentiation of rights and privileges [37,38]. Compared with the urban older, rural older people received insufficient FSS such as health care [37] and public services. Rural older people might be more dependent on family support [39]. Therefore, we divided the sample into agricultural and non-agricultural groups based on household registration status to explore the differences in the impact of ISS’s health effects between both groups. Models 7 and 8 in Table 3 indicated the grouping regression results. Care support from the son and daughter had significant negative effects within the agricultural groups, and care support from the son and spouse had significant negative effects within the non-agriculture group. Economic support and accompanied support had significant positive effects within the agriculture group. The number of intimate contacts had significantly positive effects within the non-agricultural group.

### 3.4. Moderating Effects of FSS

FSS and ISS as two aspects of social support both had significant health effects on the older, from which we could infer that there may also be a substitution effect or synergy effect between the two. Previous studies disagreed on whether there was a crowd-in [40] or crowd-out effect [41] of FSS on ISS. Therefore, we wanted to explore the moderating effect of formal support on informal support. As a major form of FSS, insurance mainly provided economic support to the older, which could reduce their family’s economic burden [42]; thus we used formal support (insurance) as a moderator variable to explore the moderating effect of FSS (insurance) in the process of economic ISS affecting the health of the older. In order to avoid the problem of collinearity, we decentralized the two variables of insurance and financial support before performing regression. Table 4 showed that the coefficient of the interaction item of economic support and insurance were −2.315 (*p* < 0.01, insurance = 0) and 0.417 (*p* < 0.05, insurance = 1), indicating that there was a significant synergy effect between economic ISS and insurance.

## 4. Discussion

This paper focused on the health effects of ISS on the older and found that different forms of ISS had different effects on older health, which was consistent with previous studies [13,43]. Care support from the son, daughter, and spouse all had negative health effects on the older [44]. Long-term care by children or spouses led to a decline in the sense of self-efficacy [44,45] and caused psychological burdens [16,17]. This psychological hint made the older put themselves in a disadvantaged position and then handed over many tasks that they were capable of doing themselves, which resulted in a lack of proper exercise. Economic support had a significantly positive health effect on the older [13]. The possible reason was that with the increase in the age of the older, physical functioning ability declined [46], the demand for medical expenses would increase with the increase in demand for medical care, and the economic support from children or others would reduce pressure on the older’s medical expenses and helped to improve older medical conditions. Accompanied support had a significantly positive health effect on the older [47,48]. Living with the older could not only provide them with some psychological comfort but also could provide some care during their daily lives [49]. Regular contact with relatives and friends could effectively promote older health [50,51]. Moreover, education and exercise both had significantly positive effects on older health. For education, there was a close connection between education and health, education was associated with better health [52,53], thus people with higher education might have better socio-economic status, higher levels of social support, and healthier lifestyle [52], which would improve older health. Moreover, people with higher education level had more extensive knowledge, they might know more health knowledge, and they would pay more attention to health maintenance. At the same time, people with higher education levels had a relatively richer cultural life, which could alleviate the individual’s mental emptiness and psychological gap after entering old age. For exercise, the elderly who often exercised had a higher health level because appropriate physical activities could not only help the elderly stretch their bones and muscles, made the body and mind smooth, but also helped reduce the prevalence of chronic diseases such as hypertension and hyperglycemia.

In addition, ISS had different health effects among different groups based on the age and household registration. Older people experienced a decline in physical functioning ability as their age increased [46] and would, therefore, face increasing medical expenses. High-age people might also be more eager to have the company and interaction of family members or others to eliminate feelings of loneliness. Compared with those who were urban, older who were rural had a disadvantaged position in terms of socio-economic status [52,54]; they also had relatively weak health awareness and insufficient FSS [37], thus they might need more ISS, especially economic support. However, we should also note that all forms of care support had significantly negative effects among the abovementioned groups, which once again showed that care support not only did not promote older health, but also had a negative impact [44]. In the context of Chinese culture, the son was the main force in supporting the older, and the daughter was only a supporting role to a certain extent. For the low-age older, their self-care level was still relatively high at this time, and they did not need too much care from their sons. However, as they grew older, the influence of traditional Chinese culture on the older made them more dependent on their sons. At this time, even if the daughters provided care support, but the absence of the son’s care would also affect the mental health and even physical health of the older. 

Furthermore, previous studies disagreed on whether there was a crowd-in [40] or crowd-out effect [41] of FSS on ISS. Using insurance as a moderating variable, our study results found that there was a significant synergy effect between economic ISS and FSS. This result was consistent with the conclusion of Hu H.W. and Luan W.J. [40]. FSS can effectively strengthen the positive health effects of ISS.

The results of this paper made certain revelations about healthy aging strategies. Firstly, the government should attach importance to and strengthen the supplemental role of ISS on FSS, and promote the effective combination of FSS and ISS, especially for older who are high-age and rural. Secondly, we should note the negative health effects of care support, which indicates that this kind of support is not strong enough and should, therefore, be further improved. In addition, families can offer care support according to the elderly’s health levels. For the elderly in good health, family members should pay attention to cultivate their independence while providing appropriate care. For the elderly with poor health or sickness, individuals can consider paying for professional nursing [11], or family members give more care support, which is not inconsistent with the conclusion that care support has a negative impact on elderly health. Government should strengthen forms of FSS such as the long-term care system, which was launched and has been operating in some pilot cities in China since 2016 [55], taking into account the psychological burden of children offering care support to the older. A government work report in 2019 proposed to “expand the pilot program of the long-term care insurance system” [56], and the National Healthcare Security Administration and Ministry of Finance of the People’s Republic of China (2020) further proposed the “Guiding Opinions on Expanding the Pilot Program of the Long-Term Care Insurance System” [57]. However, this did not mean that children did not need to bear the responsibility of caring for the older; on the contrary, this kind of care support should continue to be strengthened.

This paper also had some limitations. Firstly, according to the definition of health by WTO, health included “physical health, mental health, social adaptation, and moral health,” but we only used self-rated health to examine older health. Although self-rated health could reflect the health status of the older from both physical and psychological perspectives, if social adaptation and moral health were added to the study, the measurement of older health would be more comprehensive. In addition, older health in this paper was considered based on self-rated health and thus dependent on cognition, ability, and self-reflection, which might be more subjective. Secondly, the data used in this paper was only for the current period of 2019 thus there might be a healthiness-over-time effect. In the future, panel data with one or more periods of lag can be used for research.

## 5. Conclusions

Researching the impact of ISS on older health had important implications for the development of healthy aging policies. The present findings suggested that ISS, which included economic support, accompanied support, and the number of intimate contacts, had significant positive health effects, especially on those with high age and agricultural household registration. With respect to FSS (insurance), it could strengthen the positive health effects of economic ISS. Therefore, the government should attach importance to and strengthen the supplementary role of ISS to FSS and promote the effective combination of the two, especially for the older who are high-age and rural, and further improve the role of care support.

## Figures and Tables

**Table 1 ijerph-19-02444-t001:** Characteristics of the participants.

Variables	N (%)/M±SD	Median	Minimum	Maximum	Difference in Means
Self-rated health		3	1	5	
very good	33 (7.35)				
relatively good	160 (35.63)				
fair	196 (43.65)				
relatively poor	53 (11.80)				
very poor	7 (1.56)				
Care_son		0	0	1	0.313 (0.087) ***
No	324 (72.16)				3.441 (0.047)
Yes	125 (27.84)				3.128 (0.070)
Care_daughter		0	0	1	0.304 (0.140) *
No	410 (91.31)				3.380 (0.041)
Yes	39 (8.69)				3.077 (0.144)
Care_spouse		0	0	1	−0.086 (0.081)
No	269 (59.91)				3.320 (0.053)
Yes	180 (40.09)				3.406 (0.059)
Economic support	0.811 ± 0.237	1	0.11	1	
Accompanied support		1	0	1	−0.299 (0.114) **
No	62 (13.81)				3.097 (0.097)
Yes	387 (86.19)				3.395 (0.043)
Number of intimate contacts	12.124 ± 12.381	12.381	1	80	
Gender		1	0	1	−0.218 (0.079) **
Male	251 (55.90)				3.232 (0.060)
Female	198 (44.10)				3.450 (0.052)
Age	75.606 ± 9.446	77	60	102	
Marital status		1	0	1	−0.297 (0.082) ***
Married	293 (65.26)				3.457 (0.048)
Others	156 (34.74)				3.160 (0.067)
Child	2.503 ± 1.648	2			
Education		1	1	3	
Not attended school	87 (19.38)				0.325 (0.099) ***
Yes					3.092 (0.086)
No					3.417 (0.044)
Primary school	190 (42.32)				0.112 (0.080)
Yes					3.289 (0.064)
No					3.402 (0.050)
Middle and high school	143 (31.85)				−0.240 (0.085) **
Yes					3.517 (0.066)
No					3.278 (0.049)
University and above	29 (6.46)				−0.432 (0.160) **
Yes					3.759 (0.118)
No					3.326 (0.041)
Income		1	1	3	
Low	356 (79.29)				0.259 (0.097) **
Yes					3.301 (0.040)
No					3.559 (0.077)
Medium	69 (15.37)				−0.267 (0.109) **
Yes					3.580 (0.040)
No					3.313 (0.044)
High	24 (5.35)				−0.154 (0.176)
Yes					3.5 (0.170)
No					3.346 (0.041)
Insurance		1	0	1	−0.357 (0.422)
Yes	445 (99.11)				3.357 (0.040)
No	4 (0.89)				3 (0.408)
Smoke		1	0	1	0.180 (0.111) *
Yes	382 (85.08)				3.327 (0.044)
No	67 (14.92)				3.507 (0.094)
Drink		1	0	1	0.254 (0.104) **
Yes	371 (82.63)				3.310 (0.044)
No	78 (17.37)				3.564 (0.089)
Exercise		2	1	4	
never	129 (28.73)				0.366 (0.086) ***
Yes					3.093 (0.077)
No					3.459 (0.045)
1 to 2 times	112 (24.94)				−0.040 (0.092)
Yes					3.384 (0.076)
No					3.344 (0.046)
3 to 5 times	103 (22.94)				−0.070 (0.094)
Yes					3.408 (0.075)
No					3.338 (0.046)
6 times and above	105 (23.39)				−0.308 (0.093) ***
Yes					3.590 (0.081)
No					3.282 (0.045)

Note: (1) In the second column, we showed the number and percentage for categorical variables and the mean and standard error for continuous variables. (2) In the sixth column, for each categorical variable, we calculated the mean difference between groups, with standard errors in parentheses. (3) * *p* < 0.05, ** *p* < 0.01, *** *p* < 0.001.

**Table 2 ijerph-19-02444-t002:** Baseline results of the impact of ISS on older health.

Variables	Model 1	Model 2	Model 3	Model 4
Care_son	−0.565 ***	−0.486 ***	−0.450 ***	−0.449 ***
	(0.110)	(0.121)	(0.121)	(0.123)
Care_daughter	−0.600 ***	−0.554 **	−0.548 **	−0.572 ***
	(0.172)	(0.174)	(0.171)	(0.168)
Care_spouse	−0.313 **	−0.334 **	−0.275 *	−0.288 **
	(0.104)	(0.106)	(0.107)	(0.107)
Economic support	0.270	0.270	0.358 *	0.382 *
	(0.165)	(0.165)	(0.166)	(0.162)
Accompanied support	0.299 **	0.236	0.221	0.274 *
	(0.113)	(0.120)	(0.121)	(0.121)
Number of intimate contacts	0.006 **	0.006 *	0.006 *	0.006 *
	(0.003)	(0.003)	(0.003)	(0.003)
Gender		0.087	0.045	−0.024
		(0.085)	(0.083)	(0.089)
Age		−0.006	−0.004	0.000
		(0.004)	(0.004)	(0.005)
Marital status		0.048	−0.035	−0.042
		(0.122)	(0.126)	(0.126)
Child		0.016	0.013	0.013
		(0.022)	(0.022)	(0.022)
Education			0.150 **	0.138 **
			(0.050)	(0.038)
Income			0.064	0.074
			(0.066)	(0.066)
Insurance			0.066	0.056
			(0.332)	(0.271)
Smoke				−0.185
				(0.109)
Drink				−0.158
				(0.105)
Exercise				0.121 ***
				(0.035)
Constant	3.134 ***	3.497 ***	3.010 ***	2.703 ***
	(0.180)	(0.398)	(0.533)	(0.508)
N	449	449	449	449
R^2^	0.088	0.097	0.118	0.154
F Statistic	7.69 ***	4.94 ***	4.76 ***	5.44 ***

Note: Standard errors in parentheses; * *p* < 0.05, ** *p* < 0.01, *** *p* < 0.001. Model 1 explored the impact of care support, economic support, accompanied support, and number of intimate contacts on older health. Models 2 to 4 gradually introduced three types of control variables: personal characteristics, socio-economic characteristics and lifestyle characteristics based on Model 1.

**Table 3 ijerph-19-02444-t003:** ISS and older health: the heterogeneity by age and household registration.

	Model 5	Model 6	Model 7	Model 8
	Low-Age	High-Age	Agriculture	Non-Agriculture
Care_son	−0.493 **	−0.343	−0.404 *	−0.488 **
	(0.158)	(0.201)	(0.174)	(0.175)
Care_daughter	−0.293	−1.056 ***	−0.584 *	−0.435
	(0.193)	(0.271)	(0.230)	(0.238)
Care_spouse	−0.220	−0.322	−0.058	−0.499 ***
	(0.142)	(0.167)	(0.158)	(0.143)
Economic support	0.315	0.552 *	0.478 *	0.296
	(0.215)	(0.279)	(0.234)	(0.225)
Accompanied support	0.127	0.597 **	0.374 *	0.151
	(0.142)	(0.199)	(0.179)	(0.170)
Number of intimate contacts	0.003	0.007	0.000	0.010 *
	(0.004)	(0.004)	(0.004)	(0.004)
Control variables	Yes	Yes	Yes	Yes
Constant	2.924 ***	2.550 ***	2.584 ***	3.201 ***
	(0.561)	(0.827)	(0.758)	(0.571)
N	248	201	223	226
R^2^	0.162	0.226	0.240	0.129
F Statistic	3.41 ***	4.64 ***	4.76 ***	5.70 ***

Note: Standard errors in parentheses; * *p* < 0.05, ** *p* < 0.01, *** *p* < 0.001.

**Table 4 ijerph-19-02444-t004:** Moderating Effects of FSS on ISS.

	Coef.	Std. Err.
Economic support * Insurance		
0	−2.333 ***	0.506
1	0.392 *	0.162
Care_son	−0.443 ***	0.123
Care_daughter	−0.568 ***	0.168
Care_spouse	−0.295 **	0.107
Accompanied support	0.270 *	0.121
Number of intimate contacts	0.006 *	0.003
Insurance	−1.871 ***	0.408
Control variables	Yes	
Constant	4.615 ***	0.567
N	449	
R^2^	0.157	
F Statistic	35.73 ***	

Note: (1) 0 represented the group without insurance, and 1 represented the group with insurance. (2) * *p* < 0.05, ** *p* < 0.01, *** *p* < 0.001.

## Data Availability

The datasets used and/or analyzed during the current study are available from the corresponding author on request.

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
