# Peer review of "The Impact of Informal Social Support on Older Health: Evidence from China"

_ijerph, 2022, doi:10.3390/ijerph19042444_

Round 1

Reviewer 1 Report

Thank you for the submission of this interesting study. The paper has many strengths and is written well, but a few weak points, such as page 3, line 133, that reads 'scholars engaged with actual research'. Such statements reflect passive aggression, and the authors may wish to consider it.

There are a couple of important issues with this paper, however. First, the authors are using the term 'elderly' to refer to the participants. This term can be stigmatising, however, and associates with forced vulnerabilities and dependencies. It is a term that is not internationally used, and for the purposes of the journal that has an international audience, it may be worth rethinking the use of term. This may solve the following issue, too.

The authors refer to people aged 60 and over as 'elderly'. However, the term has been used to classify those aged 75 and over or 70 and over. People who are still productive and actively contribute to a nation's economy, for example, at the age of 60, are neither considered nor consider themselves an 'elderly' person. The more appropriate term would be 'older people' or 'older person' which I suggest the authors use.

An enriched section on the limitations of the study is needed; especially to recognise the limitations of the findings as the health levels are only considered based on self-rating, and thus dependent on cognition, ability and self-reflection. For instance, when asking 'Do you need psychological counselling?', it is most likely that those who need it will respond 'No', as studies have shown previously.

Author Response

We would like to thank you for your careful reading, helpful comments, and constructive suggestions, which has significantly improved the presentation of our manuscript.

We have carefully considered all comments from you and revised our manuscript accordingly. The manuscript has also been double-checked, and the typos and grammar errors we found have been corrected. In the following section, we summarize our responses to each comment from you.

1) page 3, line 133, that reads 'scholars engaged with actual research'. Such statements reflect passive aggression, and the authors may wish to consider it.

Response 1: Thank for your comments. We have considered it carefully and deleted "engaged with actual research" in page 3, line 131. The specific revisions are as follows:

Page 3, Line 131, Section 2.2.2

Taking the World Health Organization's definition of health as a guideline, scholars had studied older health from self-rated health (Koutsogeorgou E et al. 2015; Eric M V 2016; Liu J et al. 2019), physical health such as ADL and IADL (Liu G G et al. 2016), mental health such as degree of depression (Berkman L F et al. 1986; Zhang C and Zhang D 2016), and life satisfaction (Zeng Y and Gu D N 2002).

2) The more appropriate term would be "older people" or "older person" which I suggest the authors use.

Response 2: Thank you for pointing out this problem about using the term "elderly" to refer to the participants in manuscript. We have changed "elderly people" to "older people" in the manuscript.

3) An enriched section on the limitations of the study is needed; especially to recognise the limitations of the findings as the health levels are only considered based on self-rating, and thus dependent on cognition, ability and self-reflection.

Response 3: Thank for your above suggestion. We have enriched the limitations of the study. The specific revisions are as follows:

Page 10, line 360-367, Section 4

Firstly, according to the definition of health by WTO, health included "physical health, mental health, social adaptation and moral health", but we only use the self-rated health to examine the older health. Although self-rated health could reflect the health status of the older from both physical and psychological perspectives, if social adaptation and moral health were added to the study, the measurement of older health would be more comprehensive. In addition, older health in this paper was considered based on the self-rated health, and thus dependent on cognition, ability and self-reflection, which might be more subjective.

Reviewer 2 Report

This is a well written manuscript providing some semi=quantitative evidence for concepts that are already widely accepted, but may benefit from supportive data. As such the significance is lower but of some value.

The use of subjective survey data must be considered with caution as the are lots of opportunities for bias. This is a major limitation in the study design. Some validation with objective data is recommended, but maybe outside the scope of this manuscript.

This journal has a wide readership base. As such, it is important to define term and acronyms initially before discussing them. The authors have an unusual style. They will make a statement using jargon and/or an acronym and then define them in the subsequent paragraph.  These instances should be rewritten to define 1st and discuss after (for clarity).

More information is needed on the specific tests and parameters performed and the software used for analysis.

Overall the manuscript is well written. With some clarity in presentation, additional information on methodology and softening of the conclusions (due to the limitations of the study), this manuscript has a high potential for publication. 

Author Response

We would like to thank you for your careful reading, helpful comments, and constructive suggestions, which has significantly improved the presentation of our manuscript.

We have carefully considered all comments from you and revised our manuscript accordingly. The manuscript has also been double-checked, and the typos and grammar errors we found have been corrected. In the following section, we summarize our responses to each comment from you.

1) The use of subjective survey data must be considered with caution as the are lots of opportunities for bias. This is a major limitation in the study design. Some validation with objective data is recommended, but maybe outside the scope of this manuscript.

Response 1: Thank you for pointing out this problem in manuscript and understanding of the scope of this manuscript. We have made some appropriate statements of data reasonableness in Section 2.1. The details are as follows:

Page 3, Line 101-116, Section 2.1 Data and Sample

The data used in this paper came from a sample survey of Aging Strategy Study in Chengdu, Sichuan Province, China in 2019. This cross-sectional survey, carried out in 2019 by the Health Commission of Sichuan Province, provided valuable information on the social support and health status of the older people. In recent years, Chengdu has achieved remarkable results in healthy aging through the combination of medical care and older care. According to Chengdu Daily (2020), the proportion of the older (60 years old and above) in Chengdu in 2019 was 21.07%, and the average life expectancy was 81.01, which were higher than the national aging population proportion of 18.1% and the average life expectancy of 77.10. According to the relevant data published by the National Bureau of Statistics and National Health Commission of the People’s Republic of China, in the overall rankings of the average life expectancy of Chinese provinces in 2019, Shanghai ranked first (80.26), which was lower than Chengdu’s average life expectancy (81.01). This high life expectancy of the population indicated that Chengdu was doing a good job in offering healthy aging that conducting relevant studies on older health in Chengdu could provide a reference for the national aging strategy. Therefore, it was reasonable to select data from Chengdu in this paper.

2) This journal has a wide readership base. As such, it is important to define term and acronyms initially before discussing them. The authors have an unusual style. They will make a statement using jargon and/or an acronym and then define them in the subsequent paragraph.  These instances should be rewritten to define 1st and discuss after (for clarity).

Response 2: Thank you for pointing out this problem in manuscript. In order to simplify the content later, we only wrote the full names of informal social support and formal social support when they first appeared in the abstract and the manuscript, and then wrote them as abbreviations with a combination of initials. Specifically, informal social support was written as ISS and formal social support was written as FSS. The details can be seen as follows:

Page 1, Line 8, Abstract

To explore the impact of informal social support (ISS) on older health.

Page 1, Line 12, Abstract

and insurance was selected to explore moderating effects of formal social support (FSS)

Page 1, Line 31, Section 1 Introduction

Through Chinese cultural traditions which value family, informal social support (ISS) can provide various services to the older, especially while formal social security capacities remain insufficient; ISS provided by family members as the core has become a major determinant of older health.

Page 1, Line 35, Section 1 Introduction

Social support is a key concept of social gerontology and is closely related to older health (Rocío F B 2002). Social support can be divided into formal social support (FSS) and ISS, depending on who is providing the support.

3) More information is needed on the specific tests and parameters performed and the software used for analysis.

Response 3: Thank you for your comments. We have added some tests for the current data. First, we presented all categorical variables with the number of people and the percentage of people for each category. Second, we calculated the intergroup differences with estimated marginal means and standard errors. The specific changes can be seen in Table 1. In addition, we also added the information about Stata. The specific modifications are as follows:

Page 4, Line 197-209, Section 2.3 Statistical Analysis

Stata (version 16.0, StataCorp, Computer Resource Center, College Station, Texas, USA) is used for data analysis. Continuous variables (economic support, number of intimate contacts, age, child) are expressed as mean, standard error, median, maximum and minimum, and categorical variables (self-rated health, son’s care support, daughter’s care support, spouse’s care support, accompanied support, gender, marital status, education, income, insurance, smoke, drink, and exercise) are expressed as number, percentage, median, maximum and minimum. In addition, this paper also calculated the Intergroup differences to indicate significantly different groups with estimated marginal means and standard errors. Then, a multiple linear regression model is used as the baseline regression to explore the impact of ISS on older health; ISS serves as the independent variable and older health as the dependent variable. Furthermore, we di-vide the samples into groups according to age and household registration to explore the health effects of ISS on different groups. In this study, statistical significance is set at p<0.1.

Page 5-6, Section 3.1 Characteristics of the Participants

Table 1. Characteristics of the Participants.

Variables

N(%)/

Median

Minimum

Maximum

Difference in means

Self-rated health

3

1

5

very good

33(7.35)

relatively good

160(35.63)

fair

196(43.65)

relatively poor

53(11.80)

very poor

7(1.56)

Care_son

0

0

1

0.313(0.087) ***

No

324(72.16)

3.441(0.047)

Yes

125(27.84)

3.128(0.070)

Care_daughter

0

0

1

0.304(0.140) **

No

410(91.31)

3.380(0.041)

Yes

39(8.69)

3.077(0.144)

Care_spouse

0

0

1

-0.086(0.081)

No

269(59.91)

3.320(0.053)

Yes

180(40.09)

3.406(0.059)

Economic support

0.8110.237

1

0.11

1

Accompanied support

1

0

1

-0.299(0.114) ***

No

62(13.81)

3.097(0.097)

Yes

387(86.19)

3.395(0.043)

Number of intimate contacts

12.12412.381

12.381

1

80

Gender

1

0

1

-0.218(0.079) ***

Male

251(55.90)

3.232(0.060)

Female

198(44.10)

3.450(0.052)

Age

75.6069.446

77

60

102

Marital status

1

0

1

-0.297(0.082) ***

Married

293(65.26)

3.457(0.048)

Others

156(34.74)

3.160(0.067)

Child

2.5031.648

2

Education

1

1

3

Not attended school

87(19.38)

0.325(0.099) ***

Yes

3.092(0.086)

No

3.417(0.044)

Primary school

190(42.32)

0.112(0.080) *

Yes

3.289(0.064)

No

3.402(0.050)

Middle and high school

143(31.85)

-0.240(0.085) ***

Yes

3.517(0.066)

No

3.278(0.049)

University and above

29(6.46)

-0.432(0.160) ***

Yes

3.759(0.118)

No

3.326(0.041)

Income

1

1

3

Low

356(79.29)

0.259(0.097) ***

Yes

3.301(0.040)

No

3.559(0.077)

Medium

69(15.37)

-0.267(0.109) ***

Yes

3.580(0.040)

No

3.313(0.044)

High

24(5.35)

-0.154(0.176)

Yes

3.5(0.170)

No

3.346(0.041)

Insurance

1

0

1

-0.357(0.422)

Yes

445(99.11)

3.357(0.040)

No

4(0.89)

3(0.408)

Smoke

1

0

1

0.180(0.111) **

Yes

382(85.08)

3.327(0.044)

No

67(14.92)

3.507(0.094)

Drink

1

0

1

0.254(0.104) ***

Yes

371(82.63)

3.310(0.044)

No

78(17.37)

3.564(0.089)

Exercise

2

1

4

never

129(28.73)

0.366(0.086) ***

Yes

3.093(0.077)

No

3.459(0.045)

1 to 2 times

112(24.94)

-0.040(0.092)

Yes

3.384(0.076)

No

3.344(0.046)

3 to 5 times

103(22.94)

-0.070(0.094)

Yes

3.408(0.075)

No

3.338(0.046)

6 times and above

105(23.39)

-0.308(0.093) ***

Yes

3.590(0.081)

No

3.282(0.045)

Note: 1) In the second column, we showed the number and percentage for categorical variables, and the mean and standard error for continuous variables. 2) In the sixth column, for each categorical variable, we calculated mean difference between groups, with standard errors in parentheses. 3) * p < 0.1, ** p < 0.05, *** p < 0.01.

Reviewer 3 Report

Congratulations on taking up such socially important topic of health care for people over 60.

Unfortunately, the methods of data analysis, although very extensive, are incorrect. Were the model categorical variable, e.g. care support, introduced as categorical variable in multiple linear regression? Make sure to create a multiple linear regression model that include the categorical variable.

The conclusion "care support had negative health effects" according to the data is completely incorrect. Correctly conducted analysis should indicate who, as a primary caregiver, contributes to better health of the elderly. Other conclusions also raise serious objections.

Introducing "mental health" as part of overall health is not acceptable. Many mental illnesses occurs without the patient's awareness. Often there are people with serious mental illnesses who are not aware of it. It is impossible to objectively assess the mental health of the respondents, so please remove this aspect from the analysis.

In my opinion, it is enough to build 3 models: for Self-rated health, for ADL and for IADL and draw conclusions based on them. You can alternatively rely on two models: for Self-rated health and for objective-rated health including ADL and IADL.

In the methodology, in the additional table, please present all categorical variables with the number of people and the percentage of people for each category. Figure 1 and 2 are redundant, should be deleted. Number of children should be included in the methodology and in the model as a categorical variable. In table 1. please add the median value, as most of the data is not normally distributed.

Models for the health effect should include all variables (all from Model 4 and additionally Number of children and Insurance) excluding those variables that are highly correlated with each other. Which variables are highly correlated with each other and which will ultimately not be included in the model must be described in the methodology.

As a result of multiple linear regression model with categorical variable, the table should contain values such as: B - Unstandardized regression coefficients; PU - confidence interval; R2 c - multiple determination coefficient; eta2 -effect size; t / F - t or F Statistic; p - level of significance.

If for a categorical variable (e.g. care support) there is a statistically significant result, it is necessary to calculate Intergroup differences, indicate significantly different groups and attach a table with estimated marginal means and standard errors.

It should be more restrictive and p <0.05 should be taken as significant!

You can look for moderating effects, but only if the base model is well built.

This is not necessary, but if you want to build models that include the characteristics of the studied people, consider those variables that were important in the basic model.

In general, the basic health models themselves, when well-constructed and counted are intergroup differences in categorical variables and give the opportunity to draw conclusions. No additional calculations are needed.

I did not review the discussion because it is based on current results which are not correct.

If the data analysis is not carried out correctly, it will not be possible to draw correct conclusions and I will have to reject this work. Currently, I qualify the paper for MAJOR REVISION.

Author Response

We would like to thank you for your careful reading, helpful comments, and constructive suggestions, which has significantly improved the presentation of our manuscript.

We have carefully considered all comments from you and revised our manuscript accordingly. The manuscript has also been double-checked, and the typos and grammar errors we found have been corrected. In the following section, we summarize our responses to each comment from you.

1) The conclusion "care support had negative health effects" according to the data is completely incorrect. Correctly conducted analysis should indicate who, as a primary caregiver, contributes to better health of the elderly. Other conclusions also raise serious objections.

Response 1: Thank you for pointing out this problem in manuscript. We have modified the definition method of care support and divided it into three dummy variables, namely care_son, care_daughter, care_spouse, which indicated the care support from the son, daughter and spouse respectively. If older people are supported by them, these three dummy variables are assigned a value of 1, respectively, and 0 otherwise. The specific revisions are as follows:

Page 6-7, Section 3.2 Baseline Results

Table 3. Baseline results of the impact of ISS on older health.

Variables

Model 1

Model 2

Model 3

Model 4

Care_son

-0.565***

-0.486***

-0.437***

-0.437***

(0.110)

(0.121)

(0.121)

(0.123)

Care_daughter

-0.600***

-0.554***

-0.538***

-0.562***

(0.172)

(0.174)

(0.172)

(0.168)

Care_spouse

-0.313***

-0.334***

-0.263**

-0.275**

(0.104)

(0.106)

(0.108)

(0.107)

Economic support

0.270

0.270

0.380**

0.407**

(0.165)

(0.165)

(0.167)

(0.163)

Accompanied support

0.299***

0.236*

0.213*

0.274**

(0.113)

(0.120)

(0.124)

(0.123)

Number of intimate contacts

0.006**

0.006**

0.006**

0.006**

(0.003)

(0.003)

(0.003)

(0.003)

Gender

0.087

0.051

-0.023

(0.085)

(0.084)

(0.089)

Age

-0.006

-0.004

0.000

(0.004)

(0.004)

(0.005)

Marital status

0.048

-0.029

-0.040

(0.122)

(0.125)

(0.124)

Child

0.016

0.012

0.012

(0.022)

(0.022)

(0.022)

Education

0.105***

0.106***

(0.037)

(0.038)

Income

0.030

0.023

(0.041)

(0.040)

Insurance

0.082

0.068

(0.336)

(0.272)

Smoke

-0.195*

(0.108)

Drink

-0.162

(0.105)

Exercise

0.120***

(0.035)

Constant

3.134***

3.497***

2.931***

2.631***

(0.180)

(0.398)

(0.542)

(0.512)

N

449

449

449

449

R2

0.088

0.097

0.118

0.155

Note: Standard errors in parentheses; * p < 0.1, ** p < 0.05, *** p < 0.01. Model 1 explores the impact of care support, economic support, accompanied support, and number of intimate contacts on older health. Models 2 to 4 gradually introduce three types of control variables: personal characteristics, socioeconomic characteristics and lifestyle characteristics based on Model 1.

2) Introducing "mental health" as part of overall health is not acceptable. Many mental illnesses occurs without the patient's awareness. Often there are people with serious mental illnesses who are not aware of it. It is impossible to objectively assess the mental health of the respondents, so please remove this aspect from the analysis.

Response 2: Thank for your comments. We have removed "mental health" from the analysis, and defined older health with self-assessed health. The specific revisions are as follows:

Page 3, Line 135-141, Section 2.1 Older health

Self-rated health is an expression of social, psychological, and biological dimen-sions (Lima-Costa M F et al. 2012) which can reflect the physical and mental health status of individuals to a certain extent (Hill T D et al. 2005; Deng L Y et al. 2018). Therefore, this paper selected the self-rated health to represent the older health. It was phrased in the questionnaire as “How do you feel about your health?” The response was “very good, relatively good, fair, relatively poor, very poor”; we assigned the values 5, 4, 3, 2, 1 in order.

3) In my opinion, it is enough to build 3 models: for Self-rated health, for ADL and for IADL and draw conclusions based on them. You can alternatively rely on two models: for Self-rated health and for objective-rated health including ADL and IADL.

Response 3: Thank you for your suggestion. On this point, we would like to ask your opinion again. We think that we can choose the self-rated health for the following reasons: first, self-rated health can represent subjective and objective health; second, ADL and IADL are also both obtained by asking participants to rate a range of indicators, it is also subjective. Therefore, we think this is reasonable.

4) In the methodology, in the additional table, please present all categorical variables with the number of people and the percentage of people for each category.

Response 4: Thank you for the above suggestion. We have presented all categorical variables with the number of people and the percentage of people for each category in Table 1.

Page 5-6, Section 3.1 Characteristics of the Participants

5)If for a categorical variable (e.g. care support) there is a statistically significant result, it is necessary to calculate Intergroup differences, indicate significantly different groups and attach a table with estimated marginal means and standard errors.

Response 5: Thank you for the above suggestion. We have calculated Intergroup differences to indicate the differences of older health in different groups with estimated marginal means and standard errors. The specific revisions are as follows:

Page 5-6, Section 3.1 Characteristics of the Participants

Table 1. Characteristics of the Participants.

Variables

N(%)/

Median

Minimum

Maximum

Difference in means

Self-rated health

3

1

5

very good

33(7.35)

relatively good

160(35.63)

fair

196(43.65)

relatively poor

53(11.80)

very poor

7(1.56)

Care_son

0

0

1

0.313(0.087) ***

0

324(72.16)

1

125(27.84)

Care_daughter

0

0

1

0.304(0.140) **

0

410(91.31)

1

39(8.69)

Care_spouse

0

0

1

-0.086(0.081)

0

269(59.91)

1

180(40.09)

Economic support

0.8110.237

1

0.11

1

Accompanied support

1

0

1

-0.299(0.114) ***

0

62(13.81)

1

387(86.19)

Number of intimate contacts

12.12412.381

12.381

1

80

Gender

1

0

1

-0.218(0.079) ***

Male

251(55.90)

Female

198(44.10)

Age

75.6069.446

77

60

102

Marital status

1

0

1

-0.297(0.082) ***

Married

293(65.26)

Others

156(34.74)

Child

2.5031.648

2

Education

1

1

3

Not attended school

87(19.38)

0.325(0.099) ***

Yes

3.092(0.086)

No

3.417(0.044)

Primary school and below

190(42.32)

0.112(0.080) *

Yes

3.289(0.064)

No

3.402(0.050)

Middle and high school

143(31.85)

-0.240(0.085) ***

Yes

3.517(0.066)

No

3.278(0.049)

University and above

29(6.46)

-0.432(0.160) ***

Yes

3.759(0.118)

No

3.326(0.041)

Income

1

1

3

Low

356(79.29)

0.259(0.097) ***

Yes

3.301(0.040)

No

3.559(0.077)

Medium

69(15.37)

-0.267(0.109) ***

Yes

3.580(0.040)

No

3.313(0.044)

High

24(5.35)

-0.154(0.176)

Yes

3.5(0.170)

No

3.346(0.041)

Insurance

1

0

1

-0.357(0.422)

Yes

445(99.11)

3.357(0.040)

No

4(0.89)

3(0.408)

Smoke

1

0

1

0.180(0.111)**

Yes

382(85.08)

3.327(0.044)

No

67(14.92)

3.507(0.094)

Drink

1

0

1

0.254(0.104) ***

Yes

371(82.63)

3.310(0.044)

No

78(17.37)

3.564(0.089)

Exercise

2

1

4

never

129(28.73)

0.366(0.086) ***

Yes

3.093(0.077)

No

3.459(0.045)

1 to 2 times

112(24.94)

-0.040(0.092)

Yes

3.384(0.076)

No

3.344(0.046)

3 to 5 times

103(22.94)

-0.070(0.094)

Yes

3.408(0.075)

No

3.338(0.046)

6 times and above

105(23.39)

-0.308(0.093) ***

Yes

3.590(0.081)

No

3.282(0.045)

Note: 1) In the second column, we showed the number and percentage for categorical variables, and the mean and standard error for continuous variables. 2) In the sixth column, for each categorical variable, we calculated mean difference between groups, with standard errors in parentheses. 3) * p < 0.1, ** p < 0.05, *** p < 0.01.

6) Figure 1 and 2 are redundant, should be deleted.

Response 6: Thank for your comment. We have deleted Figure 1 and 2.

7) Number of children should be included in the methodology and in the model as a categorical variable.

Response 7: Thank for your comment. We have added the number of children (child) to the model as a control variable. The modified results are shown in Table 1.

Page 5-6, Section 3.1 Characteristics of the Participants, Table 1 (See Table 1 above)

8) In table 1. please add the median value, as most of the data is not normally distributed.

Response 8: Thank for your comment. We have added the median value in Table 1.

Page 5-6, Section 3.1 Characteristics of the Participants, Table 1 (See Table 1 above)

9) Models for the health effect should include all variables (all from Model 4 and additionally Number of children and Insurance) excluding those variables that are highly correlated with each other. Which variables are highly correlated with each other and which will ultimately not be included in the model must be described in the methodology.

Response 9: Thank for your comment. We have added all relevant variables to Model 4 in Table 2, and described in Section 2.2.4.

Page 7, Section 3.2. Baseline Results, Table 2, Model 4

Table 2. Baseline results of the impact of ISS on older health.

Variables

Model 1

Model 2

Model 3

Model 4

Care_son

-0.565***

-0.486***

-0.437***

-0.437***

(0.110)

(0.121)

(0.121)

(0.123)

Care_daughter

-0.600***

-0.554***

-0.538***

-0.562***

(0.172)

(0.174)

(0.172)

(0.168)

Care_spouse

-0.313***

-0.334***

-0.263**

-0.275**

(0.104)

(0.106)

(0.108)

(0.107)

Economic support

0.270

0.270

0.380**

0.407**

(0.165)

(0.165)

(0.167)

(0.163)

Accompanied support

0.299***

0.236*

0.213*

0.274**

(0.113)

(0.120)

(0.124)

(0.123)

Number of intimate contacts

0.006**

0.006**

0.006**

0.006**

(0.003)

(0.003)

(0.003)

(0.003)

Gender

0.087

0.051

-0.023

(0.085)

(0.084)

(0.089)

Age

-0.006

-0.004

0.000

(0.004)

(0.004)

(0.005)

Marital status

0.048

-0.029

-0.040

(0.122)

(0.125)

(0.124)

Child

0.016

0.012

0.012

(0.022)

(0.022)

(0.022)

Education

0.105***

0.106***

(0.037)

(0.038)

Income

0.030

0.023

(0.041)

(0.040)

Insurance

0.082

0.068

(0.336)

(0.272)

Smoke

-0.195*

(0.108)

Drink

-0.162

(0.105)

Exercise

0.120***

(0.035)

Constant

3.134***

3.497***

2.931***

2.631***

(0.180)

(0.398)

(0.542)

(0.512)

N

449

449

449

449

R2

0.088

0.097

0.118

0.155

F Statistic

7.69***

4.94***

4.76***

5.44***

Note: Standard errors in parentheses; * p < 0.1, ** p < 0.05, *** p < 0.01. Model 1 explores the impact of care support, economic support, accompanied support, and number of intimate contacts on older health. Models 2 to 4 gradually introduce three types of control variables: personal characteristics, socioeconomic characteristics and lifestyle characteristics based on Model 1.

10) As a result of multiple linear regression model with categorical variable, the table should contain values such as: B - Unstandardized regression coefficients; PU - confidence interval; R2 c - multiple determination coefficient; eta2 -effect size; t / F - t or F Statistic; p - level of significance.

Response 10: Thank for your comment. Due to space limitations, we only added F Statistic to the tables in the manuscript, and put the complete data in Appendix 1-3, which correspond to Tables 2-4 in the main text. Specific revisions can see the Appendix 1-3 at the end of the document.

11)You can look for moderating effects, but only if the base model is well built.

Response 11: Thank for your comment. We have rebuilt the base model (see Table 2). On the basis of the baseline regression, we again performed a moderating effect analysis, which used the insurance as the moderating variable. Specific revisions are as follows:

Page 9, Section 3.4. Moderating Effects of FSS, Table 4

Table 4. Moderating Effects of FSS on ISS.

Coef.

Std. Err.

Economic support * Insurance

0

-2.315***

0.543

1

0.417**

0.163

Care_son

-0.431***

0.123

Care_daughter

-0.557***

0.168

Care_spouse

-0.282***

0.107

Accompanied support

0.271**

0.124

Number of intimate contacts

0.006**

0.003

Insurance

-1.864***

0.445

Control variables

Yes

Constant

4.551***

0.604

N

449

R2

0.159

F Statistic

39.82***

Note: 1) 0 represented the group without insurance, and 1 represented the group with insurance. 2) * p < 0.1, ** p < 0.05, *** p < 0.01.

Appendix 1 Baseline results of the impact of ISS on older health.

Variables

Model 1

Model 2

Model 3

Model 4

B and SE

T and 95% Conf. Interval

B and SE

T and 95% Conf. Interval

B and SE

T and 95% Conf. Interval

B and SE

T and 95% Conf. Interval

Care_son

-0.565***

-5.15

-0.486***

-4.03

-0.437***

-3.61

-0.437***

-3.55

(0.110)

(-0.780, -0.349)

(0.121)

(-0.723, -0.249)

(0.121)

(-0.675, -0.199)

(0.123)

(-0.679, -0.195)

Care_daughter

-0.600***

-3.49

-0.554***

-3.18

-0.538***

-3.13

-0.562***

-3.35

(0.172)

(-0.938, -0.263)

(0.174)

(-0.897, -0.212)

(0.172)

(-0.876, -0.200)

(0.168)

(-0.892, -0.232)

Care_spouse

-0.313***

-3.01

-0.334***

-3.14

-0.263**

-2.44

-0.275**

-2.56

(0.104)

(-0.517, 0.594)

(0.106)

(-0.542, -0.125)

(0.108)

(-0.475, -0.051)

(0.107)

(-0.485, -0.064)

Economic support

0.270

1.63

0.270

1.64

0.380**

2.28

0.407**

2.49

(0.165)

(-0.055,0.594)

(0.165)

(-0.054, 0.594)

(0.167)

(0.052, 0.707)

(0.163)

(0.086, 0.728)

Accompanied support

0.299***

2.64

0.236*

1.96

0.213*

1.72

0.274**

2.22

(0.113)

(0.076, 0.521)

(0.120)

(-0.000, 0.473)

(0.124)

(-0.030, 0.456)

(0.123)

(0.031, 0.516)

Number of intimate contacts

0.006**

2.34

0.006**

2.28

0.006**

2.16

0.006**

2.11

(0.003)

(0.001, 0.012)

(0.003)

(0.001, 0.012)

(0.003)

(0.001, 0.011)

(0.003)

(0.000, 0.011)

Gender

0.087

1.03

0.051

0.61

-0.023

-0.26

(0.085)

(-0.079, 0.253)

(0.084)

(-0.113, -.215)

(0.089)

(-0.198, 0.151)

Age

-0.006

-1.34

-0.004

-0.96

0.000

0.01

(0.004)

(-0.015, 0.003)

(0.004)

(-0.013, 0.004)

(0.005)

(-0.009, 0.009)

Marital status

0.048

0.40

-0.029

-0.24

-0.040

-0.33

(0.122)

(-0.191, 0.288)

(0.125)

(-0.275, -.216)

(0.124)

(-0.285, 0.204)

Child

0.016

0.75

0.012

0.56

0.012

0.52

(0.022)

(-0.027, 0.060)

(0.022)

(-0.031, 0.056)

(0.022)

(-0.032, 0.056)

Education

0.105***

2.85

0.106***

2.81

(0.037)

(0.033, 0.178)

(0.038)

(0.032, 0.180)

Income

0.030

0.73

0.023

0.58

(0.041)

(-0.050, 0.109)

(0.040)

(-0.055, 0.101)

Insurance

0.082

0.24

0.068

0.25

(0.336)

(-0.579, -.743)

(0.272)

(-0.467, 0.604)

Smoke

-0.195*

-1.80

(0.108)

(-0.408, 0.018)

Drink

-0.162

-1.54

(0.105)

(-0.369, 0.045)

Exercise

0.120***

3.46

(0.035)

(0.052, 0.188)

Constant

3.134***

17.37

3.497***

8.79

2.931***

5.41

2.631***

5.14

(0.180)

(2.780, 3.489)

(0.398)

(2.715, 4.279)

(0.542)

(1.866, 3.996)

(0.512)

(1.624, 3.638)

N

449

449

449

449

R2

0.088

0.097

0.118

0.155

F Statistic

7.69***

4.94***

4.76***

5.44***

Appendix 2 ISS and older health: the heterogeneity by age and household registration.

Model 5

Low-age

Model 6

High-age

Model 7

Agriculture

Model 8

Non-agriculture

B and SE

T and 95% Conf. Interval

B and SE

T and 95% Conf. Interval

B and SE

T and 95% Conf. Interval

B and SE

T and 95% Conf. Interval

Care_sonCare support

-0.484***

-3.07

-0.318

-1.61

-0.368**

-2.13

-0.484***

-2.79

(0.158)

(-0.795, -0.174)

(0.198)

(-0.709, 0.073)

(0.173)

(-0.709, -0.028)

(0.173)

(-0.826, -0.142)

Care_daughter

-0.283

-1.47

-1.028***

-3.83

-0.566**

-2.49

-0.431*

 -1.82

(0.192)

(-0.663, 0.096)

(0.268)

(-1.558, -0.498)

(0.227)

(-1.014, -0.118)

(0.236)

(-0.896, 0.035)

Care_spouse

-0.216

-1.51

-0.292*

-1.76

-0.039

-0.24

-0.482***

-3.38

(0.143)

(-0.498,0.066)

(0.166)

(-0.620, 0.036)

(0.159)

(-0.352, 0.275)

(0.143)

(-0.763, -0.201)

Economic support

0.324

1.49

0.596**

2.12

0.515**

2.20

0.319

1.41

(0.217)

(-0.104, 0.752)

(0.281)

(0.042, 1.151)

(0.234)

(0.053, 0.977)

(0.227)

(-0.129, 0.767)

Accompanied support

0.140

0.97

0.600***

3.03

0.367**

2.03

0.156

0.89

(0.144)

(-0.145, 0.424)

(0.198)

(0.209, 0.991)

(0.181)

(0.010, 0.725)

(0.176)

(-0.191, 0.504)

Number of intimate contacts

0.003

0.73

0.007*

  1.82

0.001

0.19

0.010**

2.16

(0.004)

(-0.005, 0.010)

(0.004)

(-0.001, 0.015)

(0.004)

(-0.007, 0.008)

(0.004)

(0.001, 0.018)

Control variables

Yes

Yes

Yes

Yes

Constant

2.861***

5.06

2.466***

2.96

2.566***

3.33

3.120***

5.40

(0.566)

(1.746, 3.976)

(0.833)

(0.823, 4.110)

(0.771)

(1.045, 4.086)

(0.578)

(1.981, 4.259)

N

248

201

223

226

R2

0.158

0.234

0.240

0.130

F Statistic

3.27***

5.05***

4.59***

5.87***

Appendix 3 Moderating Effects of FSS on ISS.

Coef.

Std. Err.

T

95% Conf. Interval

Economic support * Insurance

0

-2.315***

0.543

-4.27

(-3.382, -1.249)

1

0.417**

0.163

2.55

(0.096, 0.738)

Care_son

-0.431***

0.123

-3.50

(-0.673, -0.189)

Care_daughter

-0.557***

0.168

-3.32

(-0.887, -0.227)

Care_spouse

-0.282***

0.107

-2.62

(-0.493, -0.071)

Accompanied support

0.271**

0.124

2.19

(0.028, 0.514)

Number of intimate contacts

0.006**

0.003

2.12

(0.000, 0.011)

Insurance

-1.864***

0.445

-4.19

(-2.740, -0.989)

Control variables

Yes

Constant

4.551***

0.604

7.53

(3.363, 5.739)

N

449

R2

0.159

F Statistic

39.82***

Round 2

Reviewer 1 Report

Thank you for this interesting submission. The paper has many strengths and addresses an important area. There may be room for expanding the discussion to make the reading more appealing to an international audience but without losing focus. 

Author Response

Dear reviewer:

We would like to thank you for your careful reading, helpful comments, and constructive suggestions, which has significantly improved the presentation of our manuscript.

We have carefully considered all comments from you and further revised our manuscript accordingly. The manuscript has also been double-checked, and the typos and grammar errors we found have been corrected. In the following section, we summarize our responses to each comment from the you. We believe that our responses have well addressed all concerns from you. We hope our revised manuscript can be accepted for publication.

In our revisions, we paid specific attention to Section 4 (Discussion).

The paper has many strengths and addresses an important area. There may be room for expanding the discussion to make the reading more appealing to an international audience but without losing focus.

Response: Thanks for your comments. We have enriched the content of Section 4 (Discussion) in the manuscript. The main revisions of Section 4 (Discussion) are as follows:

  • We further discussed the impact of control variables

Page 14, Line 479-492

And education and exercise both had significantly positive effects on older health. For education, there was a close connection between education and health, education was associated with better health (Catherine E R and Chia-ling W 1995; Lutz W and Kebede E 2018), thus people with higher education might have better socioeconomic status, higher levels of social support and healthier lifestyle (Catherine and Chia-ling 1995), which would improve older health. And people with higher education level had more extensive knowledge, they might know more health knowledge, and they would pay more attention to health maintenance; at the same time, people with higher education level had relatively richer cultural life, which could alleviate the individual's mental emptiness and psychological gap after entering old age. For exercise, the elderly who often exercised had higher health level, because appropriate physical activities could not only help the elderly stretch their bones and muscles, made the body and mind smooth, but also helped reduce the prevalence of chronic diseases such as hyper-tension, hyperglycemia.

  • For grouping regression by age, we further explained in Chinese culture context.

Page 15, Line 513-520

In the context of Chinese culture, the son was the main force in supporting the older, and the daughter was only a supporting role to a certain extent. For the low-age older, their self-care level was still relatively high at this time, and they did not need too much care from their sons. However, as they grew older, the influence of traditional Chinese culture on the older makes them more dependent on their sons. At this time, even if the daughters provided care support, but the absence of the son's care would also affect the mental health and even physical health of the older.

  • We further explained the negative coefficients of care support.

Page 15, Line 532-538

In addition, families can offer care support according to the elderly’s health levels. For elderly in good health, family members should pay attention to cultivate their independence while providing appropriate care. For the elderly with poor health or sick-ness, individuals can consider paying for professional nursing (Bonsang E 2009), or family members give more care support, which is not inconsistent with the conclusion that care support has a negative impact on elderly health.

Reviewer 3 Report

Dear Authors, you have worked very hard to prepare the manuscript and it is already much better and more readable, but still needs some corrections.

Introduction

Please write down the age at which people in China cease to be professionally active and retire. According to numerous studies, health is strongly related to professional activity.

Result section

There are different regression models, the one you chose was not the best one for the categorical variable, but the transformations you did into the care support made the multiple linear regression model applicable. I accept this analysis.

What needs improvement then? Please enhance level of significance, treat p>0.05 as not significant. Set significance levels as follows: * p <0.05, ** p <0.01, *** p <0.001.

Line 266

Add „%” at 82.63.

Line 321-344

All "p" values in the text must be checked if they are in accordance with table 2. All "-" characters must be checked if they are consistent with the table. Think about when to enter increase and when decrease, for example, if -0.565 becomes -0.437.

Discussion

It is necessary to rethink and discuss "care support had negative health effects".

The regression model shows the direction of the relationship but does not indicate what is the cause and what is the effect. In light of other ISS aspects, I suppose that those who are more ill need help from family and relatives more often, and the degree of family involvement in caring is increasing. When they are healthy, they mark "other". Think about it and discuss it. If it is true that the help of a son, daughter and relatives worsens health, does it mean that the elderly must not be helped because they will be sicker? Interpretation is very important!

Only in longitudinal studies would we be sure of causes and effects. Unfortunately, there will never be certainty here. However, based on knowledge, the most rational hypotheses should be discussed.

For me, the results in Table 3 are interesting, division by age.

What do you think, why low-age induce a significant result among sons and high-age among daughters? What is the Chinese cultural context? If this were my culture, I would consider it as the basis for a discussion about the varying degree of involvement to home care between women and men.

In my culture, it is mainly women who take care of infirm elderly, often daughters quit their jobs to look after sick parents. The sons help their parents, for example with shopping, cleaning, but usually they do not participate when it is necessary, for example, to help with personal hygiene.

It would be good to emphasize the importance and dedicate a part of discussion of other statistically significant results not related to the ISS and ESS. Education and Exercise were highly significant in Model 4.

Conclusion

After rethinking and expanding the discussion section, consider Conclusion. Certainly, the "care support had negative health effects" should be removed. Such a conclusion should not be presented to the public, because the difficulties of the infirm and elderly may even increase.

Attention. Changes made to main text should be included in the abstract.

Author Response

Dear reviewer:

We would like to thank you for your careful reading, helpful comments, and constructive suggestions, which has significantly improved the presentation of our manuscript.

We have carefully considered all comments from you and further revised our manuscript accordingly. The manuscript has also been double-checked, and the typos and grammar errors we found have been corrected. In the following section, we summarize our responses to each comment from the you. We believe that our responses have well addressed all concerns from you. We hope our revised manuscript can be accepted for publication.

In our revisions, we paid specific attention to Section 4 (Discussion), Section 5 (Conclusions).

Introduction:

1) Introduction: Please write down the age at which people in China cease to be professionally active and retire. According to numerous studies, health is strongly related to professional activity.

Response 1: Thank you for your suggestion. We have added the age at which people in China cease to be professionally active and retire. The specific revisions are as follows:

Page 1, Line 30-33

In addition, according to numerous studies, health is strongly related to professional activity (Behncke S, 2012; Grotting M W and Lillebo O S, 2020). According to China's current retirement policy, the retirement ages for most male and female in China are 60 and 55 respectively, which also poses a challenge to China's social security system.

Result section

2) Please enhance level of significance, treat p>0.05 as not significant. Set significance levels as follows: * p <0.05, ** p <0.01, *** p <0.001.

Response 2: Thanks for your comment. We have enhanced level of significance and set significance levels as follows: * p <0.05, ** p <0.01, *** p <0.001. The specific revisions are as follows:

Page 5, Line 246-247

In this study, statistical significance was set at p<0.05.

Page 7-9, Line 293

Table 1. Characteristics of the Participants.

Variables

N(%)/

Median

Minimum

Maximum

Difference in means

Self-rated health

3

1

5

very good

33(7.35)

relatively good

160(35.63)

fair

196(43.65)

relatively poor

53(11.80)

very poor

7(1.56)

Care_son

0

0

1

0.313(0.087) ***

No

324(72.16)

3.441(0.047)

Yes

125(27.84)

3.128(0.070)

Care_daughter

0

0

1

0.304(0.140) *

No

410(91.31)

3.380(0.041)

Yes

39(8.69)

3.077(0.144)

Care_spouse

0

0

1

-0.086(0.081)

No

269(59.91)

3.320(0.053)

Yes

180(40.09)

3.406(0.059)

Economic support

0.8110.237

1

0.11

1

Accompanied support

1

0

1

-0.299(0.114) **

No

62(13.81)

3.097(0.097)

Yes

387(86.19)

3.395(0.043)

Number of intimate contacts

12.12412.381

12.381

1

80

Gender

1

0

1

-0.218(0.079) **

Male

251(55.90)

3.232(0.060)

Female

198(44.10)

3.450(0.052)

Age

75.6069.446

77

60

102

Marital status

1

0

1

-0.297(0.082) ***

Married

293(65.26)

3.457(0.048)

Others

156(34.74)

3.160(0.067)

Child

2.5031.648

2

Education

1

1

3

Not attended school

87(19.38)

0.325(0.099) ***

Yes

3.092(0.086)

No

3.417(0.044)

Primary school

190(42.32)

0.112(0.080)

Yes

3.289(0.064)

No

3.402(0.050)

Middle and high school

143(31.85)

-0.240(0.085) **

Yes

3.517(0.066)

No

3.278(0.049)

University and above

29(6.46)

-0.432(0.160) **

Yes

3.759(0.118)

No

3.326(0.041)

Income

1

1

3

Low

356(79.29)

0.259(0.097) **

Yes

3.301(0.040)

No

3.559(0.077)

Medium

69(15.37)

-0.267(0.109) **

Yes

3.580(0.040)

No

3.313(0.044)

High

24(5.35)

-0.154(0.176)

Yes

3.5(0.170)

No

3.346(0.041)

Insurance

1

0

1

-0.357(0.422)

Yes

445(99.11)

3.357(0.040)

No

4(0.89)

3(0.408)

Smoke

1

0

1

0.180(0.111) *

Yes

382(85.08)

3.327(0.044)

No

67(14.92)

3.507(0.094)

Drink

1

0

1

0.254(0.104) **

Yes

371(82.63)

3.310(0.044)

No

78(17.37)

3.564(0.089)

Exercise

2

1

4

never

129(28.73)

0.366(0.086) ***

Yes

3.093(0.077)

No

3.459(0.045)

1 to 2 times

112(24.94)

-0.040(0.092)

Yes

3.384(0.076)

No

3.344(0.046)

3 to 5 times

103(22.94)

-0.070(0.094)

Yes

3.408(0.075)

No

3.338(0.046)

6 times and above

105(23.39)

-0.308(0.093) ***

Yes

3.590(0.081)

No

3.282(0.045)

Note: 1) In the second column, we showed the number and percentage for categorical variables, and the mean and standard error for continuous variables. 2) In the sixth column, for each categorical variable, we calculated mean difference between groups, with standard errors in parentheses. 3) * p < 0.05, ** p < 0.01, *** p < 0.001.

Page 10, Line 348

Table 2. Baseline results of the impact of ISS on older health.

Variables

Model 1

Model 2

Model 3

Model 4

Care_son

-0.565***

-0.486***

-0.450***

-0.449***

(0.110)

(0.121)

(0.121)

(0.123)

Care_daughter

-0.600***

-0.554**

-0.548**

-0.572***

(0.172)

(0.174)

(0.171)

(0.168)

Care_spouse

-0.313**

-0.334**

-0.275*

-0.288**

(0.104)

(0.106)

(0.107)

(0.107)

Economic support

0.270

0.270

0.358*

0.382*

(0.165)

(0.165)

(0.166)

(0.162)

Accompanied support

0.299**

0.236

0.221

0.274*

(0.113)

(0.120)

(0.121)

(0.121)

Number of intimate contacts

0.006**

0.006*

0.006*

0.006*

(0.003)

(0.003)

(0.003)

(0.003)

Gender

0.087

0.045

-0.024

(0.085)

(0.083)

(0.089)

Age

-0.006

-0.004

0.000

(0.004)

(0.004)

(0.005)

Marital status

0.048

-0.035

-0.042

(0.122)

(0.126)

(0.126)

Child

0.016

0.013

0.013

(0.022)

(0.022)

(0.022)

Education

0.150**

0.138**

(0.050)

(0.038)

Income

0.064

0.074

(0.066)

(0.066)

Insurance

0.066

0.056

(0.332)

(0.271)

Smoke

-0.185

(0.109)

Drink

-0.158

(0.105)

Exercise

0.121***

(0.035)

Constant

3.134***

3.497***

3.010***

2.703***

(0.180)

(0.398)

(0.533)

(0.508)

N

449

449

449

449

R2

0.088

0.097

0.118

0.154

F Statistic

7.69***

4.94***

4.76***

5.44***

Note: Standard errors in parentheses; * p <0.05, ** p <0.01, *** p <0.001. Model 1 explores the impact of care support, economic support, accompanied support, and number of intimate contacts on older health. Models 2 to 4 gradually introduce three types of control variables: personal characteristics, socioeconomic characteristics and lifestyle characteristics based on Model 1.

Page 12, Line 393

Table 3. ISS and older health: the heterogeneity by age and household registration.

Model 5

Model 6

Model 7

Model 8

Low-age

High-age

Agriculture

Non-agriculture

Care_son

-0.493**

-0.343

-0.404*

-0.488**

(0.158)

(0.201)

(0.174)

(0.175)

Care_daughter

-0.293

-1.056***

-0.584*

-0.435

(0.193)

(0.271)

(0.230)

(0.238)

Care_spouse

-0.220

-0.322

-0.058

-0.499***

(0.142)

(0.167)

(0.158)

(0.143)

Economic support

0.315

0.552*

0.478*

0.296

(0.215)

(0.279)

(0.234)

(0.225)

Accompanied support

0.127

0.597**

0.374*

0.151

(0.142)

(0.199)

(0.179)

(0.170)

Number of intimate contacts

0.003

0.007

0.000

0.010*

(0.004)

(0.004)

(0.004)

(0.004)

Control variables

Yes

Yes

Yes

Yes

Constant

2.924***

2.550***

2.584***

3.201***

(0.561)

(0.827)

(0.758)

(0.571)

N

248

201

223

226

R2

0.162

0.226

0.240

0.129

F Statistic

3.41***

4.64***

4.76***

5.70***

Note: Standard errors in parentheses; * p < 0.05, ** p < 0.01, *** p < 0.001.

Page 13-14, Line 453

Table 4. Moderating Effects of FSS on ISS.

Coef.

Std. Err.

Economic support * Insurance

0

-2.333***

0.506

1

0.392*

0.162

Care_son

-0.443***

0.123

Care_daughter

-0.568***

0.168

Care_spouse

-0.295**

0.107

Accompanied support

0.270*

0.121

Number of intimate contacts

0.006*

0.003

Insurance

-1.871***

0.408

Control variables

Yes

Constant

4.615***

0.567

N

449

R2

0.157

F Statistic

35.73***

Note: 1) 0 represented the group without insurance, and 1 represented the group with insurance. 2) * p < 0.05, ** p < 0.01, *** p < 0.001.

3) Line 266: Add „%” at 82.63.

Response 3: Thank you for pointing out this problem in manuscript. We have revised it and added "%" at 82.63. The specific revisions are as follows:

Page 6, Line 265

the percentage of smoking and drinking are 85.08% and 82.63% respectively.

4) Line 321-344: All "p" values in the text must be checked if they are in accordance with table 2.

Response 4: Thanks for your suggestion. We have checked all "p" values which are in accordance with table 2 in the text and revised. The specific revisions are as follows:

Page 9, Line 323-328

Model 1 showed that son’s care support (p<0.001), daughter’s care support (p<0.001), spouse’s care support (p<0.01) all had significant negative effects on older health; accompanied support (p<0.01) and number of intimate contacts (p<0.01) had a significantly positive effect on older health, but economic support had no significant impact on older health.

5) Line 321-344: All "-" characters must be checked if they are consistent with the table. Think about when to enter increase and when decrease, for example, if -0.565 becomes -0.437.

Response 5: Thanks for your suggestion. We have checked all "-" characters and they are consistent with the table. For thinking about when to enter increase and when decrease, we have added some content. The specific revisions are as follows:

Page9-10, Line 328-346

Models 2 to 4 gradually introduced three types of control variables: personal characteristics, socioeconomic characteristics and lifestyle characteristics based on Model 1. The absolute value of the son’s care support coefficient decreased from 0.565 to 0.449, the absolute value of the daughter’s care support coefficient decreased from 0.600 to 0.572, the absolute value of the spouse’s care support coefficient decreased from 0.313 to 0.288, which indicated that the negative impact of care support from the son, daughter and spouse was gradually weakening with the addition of various control variables. And the coefficient of economic support increased from 0.270 to 0.382 and became significant starting from Model 3, the coefficient of accompanied support de-creased from 0.299 to 0.221 in Model 3 and then increased to 0.274 in Model 4 which indicated that the positive effects of economic and accompanied support on older health were strengthening with the addition of various control variables. The role of ISS was becoming more and more important as the age of the older increased and eco-nomic income decreased, etc. In addition, the coefficient of the number of intimate contacts had barely changed. During this process, the suitability of fit gradually in-creased, and the conclusions remained robust.

Control variables could be seen from Model 4 in Table 2. Education (p<0.01) and exercise (p<0.001) had significant positive effects on older health. 

Discussion

6) It is necessary to rethink and discuss "care support had negative health effects".

The regression model shows the direction of the relationship but does not indicate what is the cause and what is the effect. In light of other ISS aspects, I suppose that those who are more ill need help from family and relatives more often, and the degree of family involvement in caring is increasing. When they are healthy, they mark "other". Think about it and discuss it. If it is true that the help of a son, daughter and relatives worsens health, does it mean that the elderly must not be helped because they will be sicker? Interpretation is very important!

Response 6: Thanks for your comments. Although the coefficient of current care support is significantly negative, it cannot be generalized. The specific reasons are as follows. First, as mentioned in the current discussion section (Page 14, Line 461-466), "Long-term care by children or spouses led to a decline in the sense of self-efficacy (Zimmer Z and Kwong J 2003; Wang P and Gao B 2011) and caused psychological burdens (Sabzwari S et al. 2016; Rosenberg J P et al. 2015). This psychological hint made the older put themselves in a disadvantaged position, and then handed over many tasks that they were capable of doing themselves, which results in a lack of proper exercise. " Second, the current care support provided by children, spouses, etc. in China may not be enough, so it is not enough to promote older health. However, we are very sorry that our research in this article cannot be realized, and we will further study in the follow-up. Third, our conclusions cannot be generalized either, which does not suggest that care support has a negative impact on all older people. For the older in good health, family members or relatives can provide more economic support, accompanied support, etc., while for the older in poor health, family care support is still necessary. Therefore, we have added some content in the Discussion section to further explain the negative effects of care support. The specific revisions are as follows:

Page 15, Line 532-538

In addition, families can offer care support according to the elderly’s health levels. For elderly in good health, family members should pay attention to cultivate their independence while providing appropriate care. For the elderly with poor health or sick-ness, individuals can consider paying for professional nursing (Bonsang E 2009), or family members give more care support, which is not inconsistent with the conclusion that care support has a negative impact on elderly health.

7) Only in longitudinal studies would we be sure of causes and effects. Unfortunately, there will never be certainty here. However, based on knowledge, the most rational hypotheses should be discussed.

Response 7: Thank you for pointing out this problem in manuscript and understanding of our manuscript. And we have tried our best to discuss the article clearly.

8) What is the Chinese cultural context? If this were my culture, I would consider it as the basis for a discussion about the varying degree of involvement to home care between women and men.

Response 8: Thank you for your comments. We have explained a little about the Chinese situation in the introduction (Page 2-3, Line 93-101). And we further explained the Chinese cultural context for you as follows:

Attaching importance to blood relationships and geo-relationships creates a unique community of human sentiment in China (Gao W and Chen J 2008), leading to the spread of family-centric networks that are also important for Chinese people. In addition, Chinese people pay special attention to the role of sons in supporting the older. In recent years, although the role of daughters has gradually been emphasized, the current Chinese elderly are still influenced by traditional Chinese culture and are more psychologically dependent on sons.

9) What do you think, why low-age induce a significant result among sons and high-age among daughters?

Response 9: Thank you for your comments. We have added some explanations in the Section 4 Section. The specific details are as follows:

Page 15, Line 513-520

In the context of Chinese culture, the son was the main force in supporting the older, and the daughter was only a supporting role to a certain extent. For the low-age older, their self-care level was still relatively high at this time, and they did not need too much care from their sons. However, as they grew older, the influence of traditional Chinese culture on the older makes them more dependent on their sons. At this time, even if the daughters provided care support, but the absence of the son's care would also affect the mental health and even physical health of the older.

10) It would be good to emphasize the importance and dedicate a part of discussion of other statistically significant results not related to the ISS and ESS. Education and Exercise were highly significant in Model 4.

Response10: Thanks for your suggestion. We have added the discussion about Education and Exercise in the manuscript. The specific details are as follows:

Page 14, Line 479-492

And education and exercise both had significantly positive effects on older health. For education, there was a close connection between education and health, education was associated with better health (Catherine E R and Chia-ling W 1995; Lutz W and Kebede E 2018), thus people with higher education might have better socioeconomic status, higher levels of social support and healthier lifestyle (Catherine and Chia-ling 1995), which would improve older health. And people with higher education level had more extensive knowledge, they might know more health knowledge, and they would pay more attention to health maintenance; at the same time, people with higher education level had relatively richer cultural life, which could alleviate the individual's mental emptiness and psychological gap after entering old age. For exercise, the elderly who often exercised had higher health level, because appropriate physical activities could not only help the elderly stretch their bones and muscles, made the body and mind smooth, but also helped reduce the prevalence of chronic diseases such as hyper-tension, hyperglycemia.

Conclusion

11) After rethinking and expanding the discussion section, consider Conclusion. Certainly, the "care support had negative health effects" should be removed. Such a conclusion should not be presented to the public, because the difficulties of the infirm and elderly may even increase.

Response11: Thank you for your suggestion. We have removed the "care support had negative health effects". The specific revisions are as follows:

Page 16, Line 563-566

The present findings suggested that ISS which included economic support, accompanied support and number of intimate contacts had significant positive health effects, especially on those with high age and agricultural household registration.

12) Attention. Changes made to main text should be included in the abstract.

Response12: Thanks for your comment. We have checked the abstract according to the changes made to main text and the content of the abstract did not differ from the current changes in the main text, so we have not made any changes. The specific details are as follows:

Page 1, Line 8-19

Abstract: Objectives: To explore the impact of informal social support (ISS) on older health. Methods: Multiple regression was used as the baseline regression, grouping regression was used to examine whether there were health effect differences among groups based on age and household registration, and insurance was selected to explore moderating effects of formal social support (FSS). Results: First, economic support, accompanied support and number of intimate contacts had significantly positive effects on older health except for care support’s negative effects. Second, ISS had different health effects for different groups based on age and household registration. Third, FSS was a significant moderating for ISS. Conclusions: The government should emphasize and strengthen the supplementary role of ISS to FSS and promote the effective combination of the two, especially for the older who are high-age and rural, and further improve the role of care support.

The above is all the content we have revised, and the above revised content has been revised in the manuscript. We have tried our best to revise the manuscript according to your comments and we would be very grateful if our revised manuscript can be accepted for publication.

Your kind considerations will be greatly appreciated.

With best regards,

Sincerely Yours,

Xinyuan Wang

e-mail: Veinard_wxy@outlook.com
